# Association between Obstetric Complications and Intravitreal Anti-Vascular Endothelial Growth Factor Agents or Intravitreal Corticosteroids

**DOI:** 10.3390/jpm12091374

**Published:** 2022-08-25

**Authors:** Inès Ben Ghezala, Anne-Sophie Mariet, Eric Benzenine, Marc Bardou, Alain Marie Bron, Pierre-Henry Gabrielle, Florian Baudin, Catherine Quantin, Catherine Creuzot-Garcher

**Affiliations:** 1Ophthalmology Department, University Hospital, 21079 Dijon, France; 2Inserm, CIC 1432, Clinical Investigation Center, Clinical Epidemiology/Clinical Trials Unit, University Hospital, 21079 Dijon, France; 3Biostatistics and Bioinformatics (DIM), University Hospital, 21079 Dijon, France; 4Biostatistics, Biomathematics, Pharmacoepidemiology and Infectious Diseases (B2PHI), Inserm, UVSQ, Institut Pasteur, Université Paris-Saclay, 94807 Villejuif, France; 5Gastroenterology Department, Dijon Bourgogne University Hospital, 21079 Dijon, France; 6Eye and Nutrition Research Group, CSGA, UMR1324 INRAE, 6265 CNRS, 21000 Dijon, France

**Keywords:** anti-vascular endothelial growth factor, corticosteroid, intravitreal injection, drug safety, pregnancy, obstetric complication, pharmacoepidemiology

## Abstract

This nationwide population-based cohort study aimed to describe the use of intravitreal injections (IVTs) of anti-vascular endothelial growth factor (anti-VEGF) agents and corticosteroids in pregnant women in France and to report on the incidence of obstetric and neonatal complications. All pregnant women in France who received any anti-VEGF or corticosteroid IVT during pregnancy or in the month preceding pregnancy from 1 January 2009 to 31 December 2018 were identified in the national medico-administrative databases. Between 2009 and 2018, there were 5,672,921 IVTs performed in France. Among these IVTs, 228 anti-VEGF or corticosteroid IVTs were administered to 139 women during their pregnancy or in the month preceding their pregnancy. Spontaneous abortion or the medical termination of pregnancy occurred in 10 women (16.1%) who received anti-VEGF agents and in one (3.1%) of the women who received corticosteroids (*p* = 0.09). This is the first national cohort study of pregnant women treated with anti-VEGF or corticosteroid IVTs. We found a high incidence of obstetric complications in pregnant women treated with anti-VEGF or corticosteroid IVTs but could not demonstrate a statistically significant association between the intravitreal agents and these complications. These agents should continue to be used with great caution in pregnant women.

## 1. Introduction

Intravitreal injections (IVTs) of anti-vascular endothelial growth factor (VEGF) agents have revolutionized the treatment of several retinal diseases and are widely used in developed countries today [1]. More than 1.2 million anti-VEGF IVTs were administered in France in 2020 [2]. Besides age-related macular degeneration (AMD), anti-VEGF agents are used for other indications—notably, in proliferative diabetic retinopathy [3], diabetic macular edema (DME) [4], retinal vein occlusion (RVO) [5], myopic choroidal neovascularization [6], inflammatory macular edema, inflammatory choroidopathy, and radiation maculopathy [7]. These retinal diseases may affect young people and therefore women of childbearing age. There is ongoing debate about the possible systemic adverse events following anti-VEGF IVTs. Hence, their use is not recommended during pregnancy, unless the expected benefits clearly outweigh the potential risks [8,9,10,11]. This recommendation is problematic in clinical practice, as only few alternative treatments exist. Photodynamic therapy is also not recommended during pregnancy [12]. For corticosteroid IVTs, there is no consensus regarding their use in pregnant women [13]. However, systemic corticosteroids are commonly used for pregnant women—for example, in early pregnancy or for fetal lung maturation [14]. Corticosteroid IVTs may be used for some indications such as DME or macular edema linked to RVO, but they are not indicated in all of the aforementioned diseases [13]. Untreated patients are at risk of definitive retinal or choroidal damage.

Few pharmacological data are available on the use of anti-VEGF IVTs during pregnancy. The literature mainly refers to preclinical animal studies or case series, and the data remain conflicting. A total of 29 cases of anti-VEGF IVTs during pregnancy have been documented in the literature [15,16,17,18,19,20,21,22,23,24,25,26,27,28,29,30,31,32]. The biggest case series included six women [32]. Four cases of miscarriage, one case of intrauterine fetal death due to a placental abruption, and one case of preeclampsia leading to an emergency cesarean section and neonatal complications have been reported to date [18,21,23,30,32]. However, several cases of IVTs without any obstetric complication have also been reported [20,23,24,27,28,29]. No formal conclusion can thus be drawn from these data; therefore, real-life large-scale pharmaco-epidemiological studies are needed.

Our objective was to describe the use of anti-VEGF and corticosteroid IVTs in pregnant women in France between 2009 and 2018 and to report on the incidence of obstetric and neonatal complications.

## 2. Materials and Methods

### 2.1. Study Design

This was a 10-year retrospective nationwide study conducted on the basis of the French health insurance database. This work is part of the French Epidemiology and Safety collaborative program (EPISAFE) [33].

### 2.2. Data Source

In France, all health insurance reimbursements for out-of-hospital care (visits, procedures, and drugs) are recorded in a medical–administrative database called SNIIRAM (*Système National d’Information Interrégime de l’Assurance Maladie* = French National Health Insurance Administrative Database). Moreover, administrative and medical data on every hospital stay (either in a private or public healthcare facility) are gathered in the national administrative health insurance database (*Programme de Médicalisation des Systèmes d’Information* [PMSI] = French Medical-based Information System). The PMSI is then included in the SNIIRAM. Diagnoses are encoded according to the *International Statistical Classification of Diseases and Related Health Problems, Tenth Revision* (ICD-10), and the procedures performed during the hospital stay are encoded following the *French Common Classification of Medical Procedures* (CCAM). Standardized anonymous datasets are collected from each healthcare facility and then gathered at the national level. The medical activity recorded in the PMSI directly affects the budgetary assignment of healthcare institutions. The PMSI undergoes regular monitoring. Its reliability and validity have been established in several studies [34,35].

### 2.3. Data Extraction

The National Commission for Data Protection (*Commission Nationale de l’Informatique et des Libertés no.*
*DR-2019-099*) approved the use of the SNIIRAM database, and this study adhered to the tenets of the Declaration of Helsinki. Institutional Review Board approval and patient consent were not required, as we used anonymized data from a medico-administrative database. We included all women with hospital discharge records involving a code for pregnancy (ICD-10 and CCAM codes) from 1 January 2009 to 31 December 2018 that was associated with the codes for anti-VEGF or corticosteroid IVTs during pregnancy and in the month preceding pregnancy, which were performed on an inpatient or outpatient basis. The use of anti-VEGF or corticosteroid IVTs was identified in the SNIIRAM using the CCAM code BGLB001 for IVT (“injection of a pharmacological agent in the vitreous”) and a corresponding dispensation for anti-VEGF or corticosteroid IVTs using French Presentation Identifying Codes (CIP). We included patients living in mainland France and in French overseas departments.

### 2.4. Main Outcomes

Obstetric and neonatal complications were identified using ICD-10 and CCAM codes from the records of maternal obstetric stays. The following complications were studied: pregnancy loss (early miscarriage or intrauterine fetal death), medical termination of pregnancy, hypertensive disorders of pregnancy (gestational hypertension, preeclampsia), gestational diabetes, intrauterine growth restriction, macrosomia, amniotic fluid loss or excess, fetal lesions or fetal distress, prematurity, threatened preterm labor or preterm rupture of membranes, abnormal fetal heart rate, emergency cesarean section, and neonatal distress. Data on voluntary abortions and on patient age were also collected. We searched for the indications of IVTs: diabetes, uveitis, or high myopia. We also searched for a history of hypertension prior to pregnancy, a history of diabetes prior to pregnancy, and a diagnosis of gestational diabetes, as these are well-known risk factors for obstetric complications [36,37,38]. Comparative analyses were conducted for women treated exclusively with anti-VEGF IVTs and women treated exclusively with corticosteroid IVTs. Twin pregnancies were excluded from the comparative analyses, as they were high-risk pregnancies. Comparative analyses were only performed for the 2013–2018 study period since anti-VEGF agents and corticosteroids were marginally prescribed in France before 2013. Therefore, patients treated before 2013 may not be representative of the population we aimed to study [8,10,39,40].

### 2.5. Statistical Analysis

Continuous variables with a normal distribution are expressed as means (standard deviation, SD), and categorical variables are expressed as numbers and percentages. We used a logistic regression model to assess associations between obstetric or neonatal complications and anti-VEGF IVTs, corticosteroid IVTs, age, preexisting diabetes, and preexisting hypertension. A multivariable logistic regression analysis was then performed to obtain adjusted odds ratios (ORs).

The tests were two-tailed, and statistical significance was set at *p* < 0.05. The analyses were carried out with SAS software (V.9.4.; SAS Institute).

## 3. Results

### 3.1. Descriptive Analyses

Between 1 January 2009 and 31 December 2018, a total of 5,672,921 IVTs were performed in France. Among these IVTs, 228 anti-VEGF or corticosteroid IVTs were administered to 139 women during pregnancy or in the month preceding pregnancy: 93 women had anti-VEGF IVTs only, 39 had corticosteroid IVTs only, and 7 had both anti-VEGF and corticosteroid IVTs (Table 1).

We identified 153 anti-VEGF and 75 corticosteroid IVTs. The mean number of IVTs administered during pregnancy was 1.6 ± 1.1. The distribution of IVTs administered according to the stage of the pregnancy was as follows: 23.7%, 51.8%, 19.3%, and 5.2% in the month before the onset of pregnancy, in the first trimester, in the second trimester, and in the third trimester, respectively (Table 2). 

The clinical characteristics and pregnancy outcomes of our study population are presented in Table 3. The mean age of the women was 32.5 ± 5.8 years, and 35.3% of them had diabetes.

### 3.2. Comparative Analyses

Comparative analyses were conducted for women who had anti-VEGF IVTs only and women who had corticosteroid IVTs only between 2013 and 2018, after excluding multiple pregnancies and women who received both anti-VEGF and corticosteroids IVTs (Figure 1).

Obstetric complications in 94 women were assessed after excluding women who had voluntary abortions. Spontaneous abortion or the medical termination of pregnancy occurred in 10 (16.1%) women who received anti-VEGF agents vs. 1 (3.1%) woman who received corticosteroid IVTs (*p* = 0.09). Fetal lesions or fetal distress occurred in 11 (17.7%) women who received anti-VEGF agents vs. 4 (12.5%) women who had corticosteroid IVTs (*p* = 0.51) (Table 4).

Neonatal complications in 83 women were analyzed after excluding women who had a spontaneous abortion or a medical termination of the pregnancy (Table 5).

In the multivariable analysis, anti-VEGF agents were not associated with a higher risk of obstetric and neonatal complications when compared with corticosteroids. Preexisting diabetes was associated with pregnancy hypertensive disorders (*p* = 0.002), fetal lesions or fetal distress (*p* = 0.03), an abnormal fetal heart rate (*p* = 0.048), and an emergency cesarean section (*p* < 0.001) (Table 6).

## 4. Discussion

In this study, we identified a total of 139 pregnant women who received 228 IVTs of anti-VEGF agents or corticosteroids during a 10-year period at the scale of a country of 66 million inhabitants. By comparison, 5,672,921 IVTs were performed overall in France between 2009 and 2018. IVTs in pregnant women represented 0.004% of the overall IVTs administered over the 10-year study period, which confirms that IVTs in pregnant women are very rare events. To our knowledge, this is the first nationwide study including a large number of pregnant women treated with anti-VEGF agents or corticosteroid IVTs.

We collected data on IVTs administered during pregnancy or in the month preceding pregnancy, as anti-VEGF agents can be detected in plasma until more than 20 days after their intravitreal administration [8,10,41] and could thus potentially influence early pregnancy outcomes. Interestingly, 51.8% of the IVTs reported in our study were administered in the first trimester of pregnancy, although the literature recommends being particularly cautious with anti-VEGF agents in the first trimester [25]. This may be explained by the fact that some women are not aware they are pregnant at the very beginning of their pregnancy. Some patients may thus be treated with IVTs while being unaware that they are pregnant.

We compared the incidence of obstetric and neonatal complications between pregnant women treated with anti-VEGF agents and those treated with corticosteroids, since there is widespread use of corticosteroids in pregnant women [14]. Pregnant women treated with anti-VEGF agents or corticosteroids were of similar age. Notably, a higher prevalence of diabetes was found in women treated exclusively with corticosteroids (51.3%) compared with women treated exclusively with anti-VEGF agents (26.9%), which makes sense considering the fact that corticosteroids are an alternative to anti-VEGF agents in the treatment of DME [13,39]. Only the period of 2013–2018 was included in the comparative analysis since anti-VEGF agents and corticosteroids were marginally prescribed in France before 2013. Patients treated before 2013 may not be representative of the population we aimed to study, as anti-VEGF agents were not commonly used in clinical practice (Table 1) [8,10,39,40].

No significant difference in the incidence of obstetric or neonatal complications was observed between the two groups, as was the case for the incidence of pregnancy loss or the medical termination of pregnancy, which was higher in patients treated with anti-VEGF agents compared to those treated with corticosteroids (16.1% vs. 3.1%, respectively, *p* = 0.09). This is probably related to the lack of statistical power due to the low number of patients (*n* = 62 and *n* = 32 for patients treated with anti-VEGF agents or corticosteroids, respectively). The pregnancy losses we identified in the national database were probably late miscarriages and intrauterine fetal deaths, as patients suffering from early miscarriages are rarely hospitalized. The incidence rates of pregnancy loss reported in our study are much higher than those observed in the general population of women in France, where the incidence of late miscarriage and intrauterine fetal death is less than 1% and 0.5%, respectively. Chronic hypertension and uncontrolled diabetes before pregnancy multiply the risk of intrauterine fetal death by 2.6 and 2.9, respectively [42,43].

In patients treated with anti-VEGF agents only (*n* = 93), we identified two medical terminations of pregnancy and 21 voluntary abortions. Although the reasons for these abortions were not available, the medical termination of pregnancy could have been offered to women with severe health issues triggered by pregnancy—for example, in some cases of florid diabetic retinopathy [44,45]. Some patients may also seek an abortion once they discover they are pregnant after their anti-VEGF IVT out of fear for potential fetal damage [25].

Globally, we found a high prevalence of obstetric complications in both groups: 19.4% and 31.3% of hypertensive disorders of pregnancy, 17.7% and 12.5% of fetal lesions or fetal distress, and 9.7% and 3.1% of intrauterine growth restriction in patients treated with anti-VEGF agents or corticosteroids, respectively. Pregnant women treated with anti-VEGF agents or corticosteroids had frequent comorbidities such as preexisting diabetes (26.9% and 51.3%, respectively) and preexisting hypertension (12.9% and 18.0%, respectively) and therefore more frequently had high-risk pregnancies, independently of their treatment. In the general population in France, only 0.5% of pregnant women have preexisting diabetes, and 0.7% have preexisting hypertension [46]. In our study, the average age of the women was 32.5 ± 5.8 years, which was higher than the average age of pregnant women overall in France (30.4 years old in 2016 for live births) [46]. Thus, the obstetric and neonatal complications observed in our study population may be due to the preexisting comorbidities and to the older age rather than to the IVTs. The results of our multivariable analysis support this hypothesis, since no association was found between anti-VEGF agents (vs. corticosteroids) and any obstetric or neonatal complications after adjusting for age and preexisting comorbidities. However, the confidence intervals were very wide due to our small sample size. We found an association between preexisting diabetes and several obstetric complications, which is consistent with the literature (Table 6) [37]. Moreover, some ophthalmologic complications such as DME or proliferative diabetic retinopathy are related to poor diabetes control [47]. Pregnant women with uncontrolled diabetes are at risk of DME and therefore have even more high-risk pregnancies than women with well-controlled diabetes [37].

Our study has some limitations. First, despite a 10-year study period, the number of patients studied was quite small (*n =* 139 in the descriptive analysis, *n* = 94 in the comparative analysis), limiting the power to reach statistical significance for most of our comparisons. However, it must be emphasized that corticosteroids and anti-VEGF IVTs in pregnant women are very rare events, as they represented 0.004% of the overall IVTs over the 10-year study period. Second, we lacked clinical data on the pregnant women included in our study. Some clinical factors that are well-known risk factors for obstetric complications could not be confidently identified in the database and included in our analysis—for example, smoking and obesity. Indeed, even if smoking, for instance, can be registered, the accuracy of the recording is quite low. Regarding obesity, body mass index is insufficiently documented in the PMSI before pregnancy, and body mass index is not reliable during pregnancy.

Nonetheless, we were able to identify some important clinical factors involved in obstetric complications, that is, maternal age, diabetes, and hypertension. Moreover, because data on neonatal stays were not available in our study, neonatal complications were studied only on the basis of maternal stays, which could lead to an underestimation of neonatal complications. Third, because of the very small number of women available for inclusion, we were not able to study the effect of the different anti-VEGF agents administered (bevacizumab, ranibizumab, or aflibercept) on obstetric complications, although they differ with regard to their pharmacokinetic and pharmacodynamic parameters [48]. Interestingly, bevacizumab and ranibizumab only bind to VEGF, whereas aflibercept also binds to the placental growth factor (PlGF). PlGF may play a role in embryo development and implantation and in fetoplacental circulation [49]. Finally, corticosteroids were used as a reference group since they are widely used in pregnant women, especially systemic corticosteroids. However, there is controversy regarding the potential adverse events—notably, teratogenicity, a reduction in birth size, and cerebral palsy—and IVT corticosteroids should not be considered a totally harmless treatment [14,50].

Our study also has several strengths. It was a national study over a 10-year period using the medico-administrative database. IVTs in pregnant women are extremely rare, as we have demonstrated. Other studies have shown that the national medico-administrative database is suitable for studying rare events such as postoperative endophthalmitis [35,51].

## 5. Conclusions

The use of the medico-administrative database allowed us to establish an exhaustive collection of data on pregnant women treated with IVTs of anti-VEGF agents or corticosteroids during a 10-year study period at a national level. We reported on all the cases (139) of pregnant women treated with anti-VEGF agents or corticosteroid IVTs during pregnancy or in the month before pregnancy in France between 2009 and 2018; our findings confirmed that IVTs in pregnant women are rare events. Even though we found a high incidence of obstetric complications in our series, we could not demonstrate a statistically significant association between the intravitreal agents and these complications after adjusting for age and preexisting comorbidities. Indeed, this high rate of obstetric complications is likely supported by the underlying condition responsible for the IVT treatment. Nevertheless, due to the potential lack of statistical power, it is impossible to exclude a possible relationship between these treatments and some of the obstetric complications observed. Anti-VEGF agents and corticosteroids delivered through the intravitreal route should continue to be used with great caution in pregnant women and in women of childbearing age after carefully weighing the benefits and the potential risks.

## Figures and Tables

**Figure 1 jpm-12-01374-f001:**
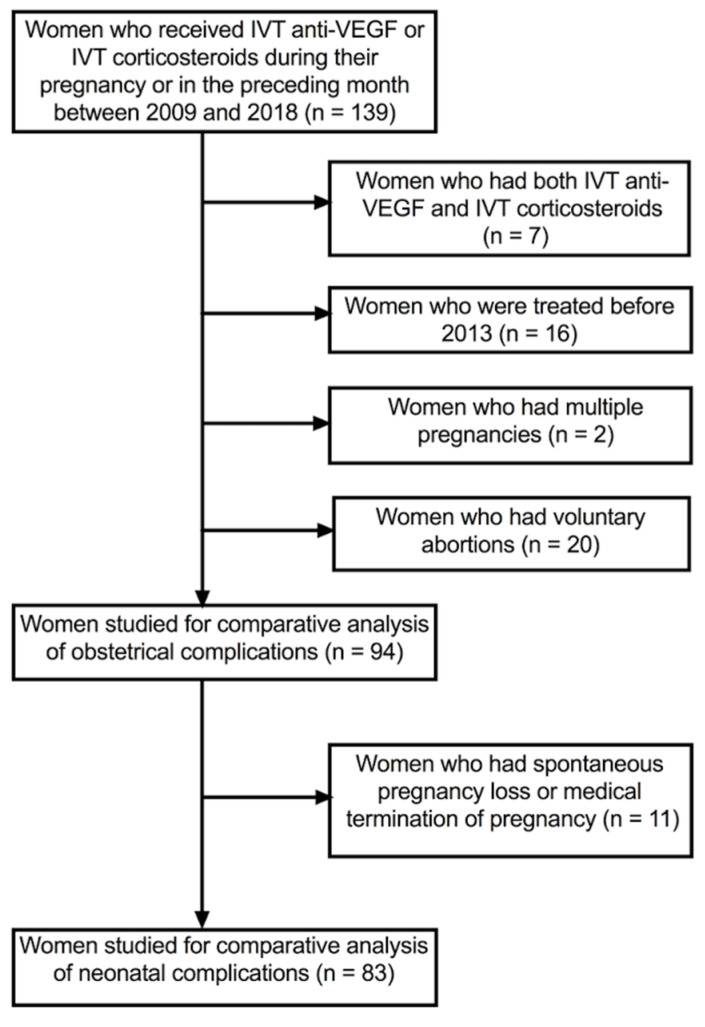
Flowchart of the study.

**Table 1 jpm-12-01374-t001:** Number of pregnant women treated with anti-VEGF or corticosteroid IVTs in France from January 2009 to December 2018.

	2009	2010	2011	2012	2013	2014	2015	2016	2017	2018	Total
Anti-VEGF agents only	2	4	2	4	5	8	17	15	10	26	93
Corticosteroids only	1	2	0	1	4	4	9	6	5	7	39
Anti-VEGF agents and corticosteroids	0	0	0	0	1	1	3	1	1	0	7

Anti-VEGF, anti-vascular endothelial growth factor; IVTs, intravitreal injections.

**Table 2 jpm-12-01374-t002:** Number of intravitreal injections administered in France from January 2009 to December 2018, according to the stage of pregnancy.

	Month before Onset of Pregnancy	First Trimester	Second Trimester	Third Trimester	Total
Anti-VEGF agents	47	82	18	6	153
Corticosteroids	7	36	26	6	75

Anti-VEGF, anti-vascular endothelial growth factor.

**Table 3 jpm-12-01374-t003:** Clinical characteristics and pregnancy outcomes of women who had anti-VEGF or corticosteroid IVTs during pregnancy or in the month preceding pregnancy between 2009 and 2018 (*n* = 139).

	Anti-VEGF Agents Only (*n* = 93)	Corticosteroids Only (*n* = 39)	Anti-VEGF Agents and Corticosteroids (*n* = 7)
Age, years	33.1 ± 5.8	31.2 ± 5.9	32.3 ± 4.1
Medical history			
Preexisting diabetes	25 (26.9%)	20 (51.3%)	4 (57.1%)
Preexisting hypertension	12 (12.9%)	7 (18.0%)	2 (28.6%)
Uveitis	2 (2.2%)	9 (23.1%)	1 (14.3%)
High myopia	6 (6.5%)	1 (2.6%)	0 (0.0%)
Single or multiple pregnancy			
Single pregnancy	91 (97.9%)	39 (100.0%)	7 (100.0%)
Twin pregnancy	2 (2.2%)	0 (0.0%)	0 (0.0%)
Pregnancy outcome			
Voluntary abortion	21 (22.6%)	5 (12.8%)	0 (0.0%)
Pregnancy loss	10 (10.8%)	1 (2.6%)	0 (0.0%)
Medical termination of pregnancy	2 (2.1%)	0 (0.0%)	0 (0.0%)
Live birth	60 (64.5%)	33 (84.6%)	7 (100.0%)

Anti-VEGF, anti-vascular endothelial growth factor; IVTs, intravitreal injections.

**Table 4 jpm-12-01374-t004:** Comparison of obstetric complications and pregnancy outcomes between women who had anti-VEGF agents only vs. corticosteroid IVTs only during pregnancy or in the month preceding pregnancy, between 2013 and 2018 (*n* = 94).

	Anti-VEGF Agents Only (*n* = 62)	Corticosteroids Only (*n* = 32)	*p*
Obstetric complications			
Gestational diabetes	6 (9.7%)	6 (18.8%)	0.33
Hypertensive disorders of pregnancy	12 (19.4%)	10 (31.3%)	0.20
Fetal lesions or fetal distress	11 (17.7%)	4 (12.5%)	0.51
Intrauterine growth restriction	6 (9.7%)	1 (3.1%)	0.42
Macrosomia	4 (6.5%)	2 (6.3%)	0.99
Amniotic fluid loss or excess	2 (3.2%)	1 (3.1%)	0.99
Pregnancy outcome			
Pregnancy loss or medical termination of pregnancy	10 (16.1%)	1 (3.1%)	0.09
Live birth	52 (83.9%)	31 (96.9%)	

Anti-VEGF, anti-vascular endothelial growth factor; IVTs, intravitreal injections. Patients who had voluntary abortions were excluded from the analysis.

**Table 5 jpm-12-01374-t005:** Comparison of neonatal complications and pregnancy outcomes between women who had anti-VEGF IVTs only vs. women who had corticosteroid IVTs only during pregnancy or in the month preceding pregnancy, between 2013 and 2018 (*n* = 83).

	Anti-VEGF Agents Only (*n* = 52)	Corticosteroids Only (*n* = 31)	*p*
Abnormal fetal heart rate	15 (28.9%)	7 (22.6%)	0.53
Neonatal distress	2 (3.9%)	1 (3.2%)	0.99
Prematurity vs. full-term pregnancy			0.33 *
Extreme and very preterm	1 (1.9%)	1 (3.2%)	
Moderate	4 (7.7%)	0 (0.0%)	
Late preterm	7 (13.5%)	7 (22.6%)	
Full-term pregnancy	40 (76.9%)	23 (74.2%)	
Emergency cesarean section	19 (36.5%)	12 (38.7%)	0.84
Threatened preterm labor or preterm rupture of membranes	3 (5.8%)	5 (16.1%)	0.14

Anti-VEGF, anti-vascular endothelial growth factor; IVTs, intravitreal injections. Patients who had a voluntary abortion, spontaneous pregnancy loss, or medical termination of the pregnancy were excluded from the analysis. * Global test for all prematurity levels.

**Table 6 jpm-12-01374-t006:** Multivariable analysis of factors associated with obstetric and neonatal complications in women who had anti-VEGF IVTs only vs. women who had corticosteroid IVTs only during pregnancy or in the month preceding pregnancy, between 2013 and 2018.

	Hypertensive Disorders of Pregnancy *		Fetal Lesi Ons or Fetal Distress *		Intrauterine Growth Restriction *	
	OR (95% CI) *	*p*	OR (95% CI) *	*p*	OR (95% CI) *	*p*
Anti-VEGF agents vs. corticosteroids	0.75 (0.22–2.51)	0.64	2.05 (0.56–7.55)	0.28	4.53 (0.49–42.16)	0.18
Age ≥ 35	0.50 (0.13–1.88)	0.31	1.78 (0.56–5.67)	0.33	2.75 (0.55–13.78)	0.22
Preexisting hypertension	8.38 (2.16–32.50)	0.002	-	-	-	-
Preexisting diabetes	7.01 (2.06–23.90)	0.002	3.68 (1.12–12.07)	0.03	3.36 (0.65–17.25)	0.15
	**Pregnancy loss or medical termination of pregnancy ***		**Abnormal fetal heart rate ****		**Emergency cesarean section ****	
	OR (95% CI)	*p*	OR (95% CI)	*p*	OR (95% CI)	*p*
Anti-VEGF agents vs. corticosteroids	5.99 (0.72–49.84)	0.10	1.75 (0.57–5.32)	0.33	2.04 (0.61–6.82)	0.25
Age ≥ 35	2.50 (0.67–9.29)	0.17	0.68 (0.23–2.07)	0.50	1.99 (0.64–6.21)	0.23
Preexisting hypertension	0.39 (0.04–3.49)	0.40	0.37 (0.08–1.65)	0.19	1.33 (0.33–5.46)	0.69
Preexisting diabetes	-	-	3.15 (1.01–9.80)	0.048	14.09 (4.04–49.18)	<0.001

Anti-VEGF, anti-vascular endothelial growth factor; IVTs, intravitreal injections. -, not included in the multivariable analysis. * *n* = 94. Patients who had voluntary abortions were excluded from the analysis. ** *n* = 83. Patients who had a voluntary abortion, spontaneous pregnancy loss, or medical termination of the pregnancy were excluded from the analysis.

## Data Availability

The SNIIRAM and PMSI databases were made available by the French National Agency for the Management of Hospitalization Data (Agence technique de l’information sur l’hospitalisation, ATIH). The use of these data by our department was approved by the CNIL. We are not permitted to share these data. SNIIRAM and PMSI data are available from ATIH to researchers who meet the criteria for access (requests for access are evaluated by the CNIL).

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
