# Peer review of "Association between Obstetric Complications and Intravitreal Anti-Vascular Endothelial Growth Factor Agents or Intravitreal Corticosteroids"

_jpm, 2022, doi:10.3390/jpm12091374_

Round 1
Reviewer 1 Report
I congratulate the authors on doing this study on the effects of intravitreal treatment with anti-VEGF Vs corticosteroids and assessing the complications by using a national dataset. The increase in use of these agents in treatment of the conditions which are becoming more prevalent makes this study very relevant. moreover, the study is well written. The fact that this is a rare complication and intervention, even using a national database over few years study period didnt provide a well powered study. This definitely has its limitations in coming at a conclusion which authors have duly noted. We are aware that both agents are associated with poor neurodevelopmental outcome when used in pregnancy and neonatal age. There are studies showing that use of steroids in pregnancy for threatened preterm delivery which eventually go on till term gestation age are associated with poor head growth. similarly use of anti-VEGF (Bevacuzimab) in neonates with retinopathy of prematurity has shown to have poor neurodevelopmental outcome, whether this is because of the severity of disease or the therapeutic agent itself is not well delineated.
So one would expect a similar outcome when used during pregnancy. The study clearly shows increase in complications but as the authors stated this could all be due to the disease process itself in these mothers. Moreover use of a national database may not be exhaustive of all the complications. Also the neonatal complication after delivery like hospital stay is not ascertained in this study.
Being a rare intervention we will have to use the national database based studies as a prospective study when the incidence is so low is not possible. and the authors have done a good job about that. Whether it adds anything to the current knowledge is a question that is not well answered
Author Response
I congratulate the authors on doing this study on the effects of intravitreal treatment with anti-VEGF Vs corticosteroids and assessing the complications by using a national dataset. The increase in use of these agents in treatment of the conditions which are becoming more prevalent makes this study very relevant. moreover, the study is well written. The fact that this is a rare complication and intervention, even using a national database over few years study period didnt provide a well powered study. This definitely has its limitations in coming at a conclusion which authors have duly noted.
We thank the Reviewer for their comments.
We are aware that both agents are associated with poor neurodevelopmental outcome when used in pregnancy and neonatal age. There are studies showing that use of steroids in pregnancy for threatened preterm delivery which eventually go on till term gestation age are associated with poor head growth. similarly use of anti-VEGF (Bevacuzimab) in neonates with retinopathy of prematurity has shown to have poor neurodevelopmental outcome, whether this is because of the severity of disease or the therapeutic agent itself is not well delineated.
We updated our discussion and our references to mention the potential association between antenatal corticosteroids and reduction in birth size (p.9 lines 315-316).
So one would expect a similar outcome when used during pregnancy. The study clearly shows increase in complications but as the authors stated this could all be due to the disease process itself in these mothers. Moreover use of a national database may not be exhaustive of all the complications. Also the neonatal complication after delivery like hospital stay is not ascertained in this study.
We mentioned this potential limitation in our discussion (p 9 lines 303-306).
Being a rare intervention we will have to use the national database based studies as a prospective study when the incidence is so low is not possible. and the authors have done a good job about that. Whether it adds anything to the current knowledge is a question that is not well answered
We thank the Reviewer for their comments.
Reviewer 2 Report
The article is dedicated to a very important topic, real-life large-scale pharmaco-epidemiological study of anti-VEGF and corticosteroid IVTs during pregnancy. The question of anti-VEGF and corticosteroid IVTs during pregnancy causes hot debates and has controversial data, so real-life unbiased impersonal research with distinct results is very much needed.
The manuscript carries high content merit. Statistical tools are adequately applied, the results do not cause doubts by figures. Limitations of the study are well described in the discussion section. The article is well-designed and clearly written.
Tiny comments include figuration suggestions for this manuscript:
1. It is better to move table 1 and table 6 in the text so it is on one page, not two.
2. Table 2 corticosteroid(s) - it is better to make the word in one line, not two.
3. Table 3 last column name - it's better to make it in 3 lines, not 4
In table 5 the question about p value is in the picture attached.

Author Response
The article is dedicated to a very important topic, real-life large-scale pharmaco-epidemiological study of anti-VEGF and corticosteroid IVTs during pregnancy. The question of anti-VEGF and corticosteroid IVTs during pregnancy causes hot debates and has controversial data, so real-life unbiased impersonal research with distinct results is very much needed.
The manuscript carries high content merit. Statistical tools are adequately applied, the results do not cause doubts by figures. Limitations of the study are well described in the discussion section. The article is well-designed and clearly written.
We thank the Reviewer for their comments.
Tiny comments include figuration suggestions for this manuscript:
1. It is better to move table 1 and table 6 in the text so it is on one page, not two.
2. Table 2 corticosteroid(s) - it is better to make the word in one line, not two.
3. Table 3 last column name - it's better to make it in 3 lines, not 4
We thank the Reviewer for their remarks. The manuscript was reformatted by the Editors, that is why some tables were split on two pages. We updated the manuscript accordingly.
In table 5 the question about p value is in the picture attached.
The p-value refers to the global test of prematurity (including all different prematurity levels). We mentioned this clearly in the table legend.